

# Stratigraphic Identification with Airborne Electromagnetic Methods at the Hanford Site, Washington

Piyoosh Jaysaval, Judith L. Robinson, and Timothy C. Johnson

Pacific Northwest National Laboratory, 902 Battelle Blvd, Richland, WA 99354, USA

*Correspondence:* Piyoosh Jaysaval (piyoosh.jaysaval@pnnl.gov)

**Abstract.** Stratigraphic units can influence the fate and transport of subsurface contaminants within groundwater. Units having coarse-grained sediments act as preferential flow pathways, and therefore can accelerate the transport of contaminants to reach human and ecological receptors. At legacy waste sites, detailed knowledge of subsurface stratigraphy can be used for effective monitoring and remediation

planning to help minimize risk to human health and the environment. Airborne electromagnetic (AEM) methods can non-invasively provide information on kilometer-scale or larger subsurface stratigraphic features and fill informational gaps in directly sampled data from sparsely located boreholes. In this paper, we present inversion results of a 412 line-km frequency-domain AEM survey to delineate subsurface stratigraphic features at the Hanford Site, located in southeastern Washington State. The inversion was

performed using a massively parallel 3D electromagnetic modeling and inversion code, where the modeling is based on solving frequency-domain Maxwell's equations using an unstructured-mesh finite-element method and the inversion employs a Gauss-Newton optimization scheme. The results are compared to an underlying geologic framework model (GFM), built by interpolating contact depths of stratigraphic units interpreted from site borehole datasets. In areas with good borehole coverage, the inversion results show a

good match with the GFM to a depth of about 60 m. Outside of these areas, the inversion results exhibit inconsistencies from the assumptions made to create the GFM, demonstrating that the AEM survey results can be used to improve the understanding of the geological conceptual model.



## 1. Introduction

Subsurface stratigraphy is often mapped by interpolating geologic contact depths from borehole datasets. Near boreholes, geologic contacts are presumably more accurate than estimates at greater distances. Interpolation of these depths may not be reliable at places where boreholes are sparsely located, or where subsurface heterogeneity exists at length scales smaller than the distance between boreholes. Improved methods are needed to provide accurate spatial extent of subsurface stratigraphic units. Surface and/or airborne geophysical methods — e.g., electrical resistivity tomography (ERT), active source electromagnetic (EM) methods, seismic methods, ground penetrating radar, and magnetic methods — are obvious choices that can provide high-resolution images of the subsurface covering large areas. For kilometer-scale and larger investigations, airborne EM (AEM) methods are particularly cost-effective ways to obtain subsurface coverage, with minimum disturbance, in a relatively short period of time.

There are primarily two types of AEM methods: frequency-domain and time-domain — detailed capabilities and differences of each method are presented in Siemon et al. (2009a), Steuer et al. (2009), and Thomson et al. (2007). In particular, the frequency-domain AEM method is more suitable for high-resolution investigations at shallower depths, while the time-domain AEM method is suitable for very large coverage areas and deeper exploration depths (Siemon et al., 2009a). Both AEM methods have been used throughout the world for mapping of geoelectrical structures in the near surface, at depths down to a few hundred meters, for the exploration of minerals (Christensen et al., 2009; Smith, 2010; Sykes et al., 2006), groundwater (Sattel and Kgotlhang, 2004; Schamper et al., 2013; Siemon et al., 2011), and oil and gas (Cox et al., 2012; Pfaffhuber et al., 2009; Smith, 2010; Smith et al., 2008; Walker and Rudd, 2008); and for environmental, hazard and engineering investigations (Finn et al., 2007; Goebel et al., 2019; Minsley et al., 2012; Pfaffhuber et al., 2017). These applications utilize the fact that the interpreted electrical resistivity (or conductivity) anomalies can directly be correlated with subsurface properties. For example, in sedimentary rocks, the resistivity tends to decrease as the grain size decreases and as clay content increases; unsaturated sequences will have a higher resistivity value than their saturated counterparts; hydrocarbon saturation in pore space tends to increase resistivity compared to water saturation (Palacky, 1987).

As AEM surveys generate a large amount of data consisting of tens of thousands of sounding locations, data are normally interpreted using 1D analyses, e.g., by applying apparent resistivity transforms (Huang and Fraser, 1996; Sengpiel, 1988), conductivity-depth imaging (Huang and Rudd, 2008; Macnae et al., 1991), 1D inversions (Chen and Raiche, 1998; Farquharson et al., 2003), or laterally/spatially constrained 1D inversion (Auken and Christiansen, 2004; Siemon et al., 2009b; Viezzoli et al., 2008). Despite widespread acceptance and the argument that spatially constrained 1D inversion can provide good results


for geological settings where each observation can be simulated with 1D forward modeling (Auken et al., 2008; Viezzoli et al., 2010), it has been demonstrated that such 1D analyses are usually not adequate when 2D or 3D geological complexity exists (e.g., see Cox et al., 2010; Ellis, 1998; Raiche et al., 2001; Wilson et al., 2006). Several 2D and 3D modeling and inversion analyses have also been developed (e.g., by Cox et al., 2010; Heagy et al., 2017; Noh et al., 2018; Sasaki, 2001; Xi and Li, 2016; Yang et al., 2014); however, their acceptance has been limited because the number of observations used in the inversion was restricted due to computational constraints, or because the inversion codes were not publicly available.

In 2008, frequency-domain AEM data were acquired in the eastern part of the Hanford Site to delineate highly transmissive zones that may serve as preferential flow pathways for subsurface contaminants. Initial interpretation of the data was based on a 1D analysis by transforming the flight-line data into depth slices of electrical resistivity by computing apparent resistivity and apparent depth (Huang and Fraser, 1996). This transformation showed a weak correlation of resistivity with the underlying stratigraphy derived from nearby boreholes, which is likely the result of representing the subsurface from interpolation of simplified 1D analysis as opposed to a 2D or 3D analysis. In 2010, two ground-based geophysical surveys, ERT and frequency-domain ground-based EM, were conducted along a test profile adjacent to a portion of the AEM survey flight lines, with the main objective of validating the AEM data interpretation (CHPRC, 2010a, 2010b). Both ERT and ground-based EM data were inverted using 2D inversion schemes and the obtained resistivity models were fairly consistent with the stratigraphic units interpreted from six nearby borehole datasets down to 30–40 m depth. However, these inversion results showed poor agreement with the resistivity slices computed from the AEM data.

In this paper, we present the first inversion results of the AEM data at the Hanford Site obtained using a 3D EM modeling and inversion code. This code has recently been developed as a part of a capability extension to E4D, an open-source software widely used for 3D/4D ERT data modeling and inversion (Johnson et al., 2010, 2017; Johnson and Wellman, 2015). The forward modeling is based on solving frequency-domain Maxwell's equations using an unstructured-mesh finite-element method, and the inversion uses a Gauss-Newton optimization scheme. The implementation is massively parallelized, which allows for forward and inverse simulations to be efficiently distributed on thousands or even more processing cores of a supercomputer for large-scale problems. For survey configurations similar to the collected AEM data, a synthetic inversion study was first executed to determine if a full 3D inversion could recover the subsurface resistivity between flight paths. Inversions were then performed along individual flight lines because the synthetic study demonstrated that the flight-line spacing was too large for a full 3D inversion. The inversion results were subsequently compared to an existing geologic conceptual model developed by interpolating contact depths of stratigraphic units interpreted from site borehole datasets.



The remainder of this paper is organized as follows: we first present a brief description of our study area, the Hanford Site. We then describe our methodology with details on the AEM system and data acquisition, 3D EM modeling and inversion code, and relationship of electrical resistivity with different stratigraphic units at the Hanford Site. We thereafter present inversion results of the AEM data, followed by their interpretation. Finally, we draw concluding remarks on our work.

## 2. Study area

The Hanford Site is a part of the U.S. Department of Energy (DOE) nuclear weapons complex. It is located in southeastern Washington State (Fig. 1) and covers an area of about 1505 km$^2$. The Columbia River passes through the north portion and surrounds the south-eastern quadrant of the Site. In 1943, the U.S. federal government acquired the Hanford Site to build reactors for a mission to produce plutonium for nuclear weapons and perform research on plutonium production. Between 1943 and 1963, nine reactors were built to produce weapons-grade plutonium from irradiated uranium. By 1987, these reactors produced nearly two-thirds of the plutonium used in the United States. During this operational period, the chemical processing plants processed about 110,000 metric tons of irradiated fuel from the reactors. This led to discharge of an estimated 450 billion gallons of contaminated liquids to soil through cribs, ponds, and ditches, and about 56 million gallons of highly radioactive wastes to 177 giant underground (149 single-shell and 28 double-shell) tanks — mostly at the center of the Hanford Site (the Central Plateau, marked with a red dashed line in Fig. 1) (Gephart, 2003; Lichtenstein, 2004). According to published historical reports on the Hanford Site, 67 single-shell tanks have, or are suspected to have, collectively leaked over 1 million gallons of highly radioactive wastes into the vadose zone (Gephart and Lundgren, 1998; Gephart, 2003; Lichtenstein, 2004).

The hazardous wastes presented a risk of contaminating groundwater and possibly the nearby Columbia River through subsurface flow pathways. A recent groundwater monitoring report at the Hanford Site reported that about 168 km$^2$ of groundwater is contaminated above regulatory standards (DOE, 2019). Groundwater across the Hanford Site typically flows from west to east and discharges along the Columbia River. Stratigraphic units and paleochannels, filled with coarse-grained sediments, act as preferential flow pathways to contaminants within groundwater, and consequently can shorten travel times to reach human and ecological receptors. It has been estimated that the contaminant travel time from the Central Plateau to the Columbia River can vary from a few decades for the most mobile contaminants to a century or more for the less mobile ones within the groundwater (Gephart, 2003). For effective monitoring and remediation





planning, it is therefore critical to have an accurate understanding of the stratigraphic units and possible flow pathways that can influence the subsurface contaminant transport across the Hanford Site.

The generalized stratigraphic units of the Hanford Site consist of, in descending order, the Hanford formation, the Ringold Formation, and the Columbia River Basalt Group (Fig. 2). This simplified layer-cake depositional model has been complicated by the depositional process, and subsequent removal, of sedimentary units. During the Ringold Formation depositional cycle, actions of ancestral rivers resulted in deposition of intercalated layers of indurated to semi-indurated sediment including clay, silt, fine-to coarse-grained sand, and granule-to-cobble gravel. Within most of the Ringold Formation, there is a thick confining layer of lower mud, called the Ringold Mud unit, composed of lacustrine silts and clay, paleosol, and fluvial overbank. After the Ringold Formation deposition, regional incision occurred for an extended period, followed by soil formation and deposition of windblown sediments. These sedimentary deposits are termed the Cold Creek unit. This was followed by the deposition of coarse gravel-to-boulder sediments of the Hanford formation due to cataclysmic floods during the last ice age. The floods also incised channels of different depth into the previously deposited Ringold and Cold Creek sediments, and in some cases, probably the basaltic bedrock. These scoured areas and paleochannels were later filled by poorly sorted Hanford sands and gravels. The above descriptions are gathered from DOE (2011); however, for comprehensive details on the stratigraphy at the Hanford Site, interested readers are further referred to DOE (2002), Lindsey (1995), and Lindsey et al. (1994).

Across the Hanford Site, these stratigraphic units have continuously been mapped to build a 3D geologic framework model (GFM) by interpolating depths of different geologic contacts interpreted from approximately 1500 boreholes (CHPRC, 2018). Borehole datasets may include drillers' descriptions, geologists' descriptions, geophysical logs, grain-size analyses, and/or sediment photographs, and were also used to establish the presence of ancestral fluvial channels or paleochannels across the site (Bjornstad et al., 2010; Reidel and Chamness, 2007). The 3D GFM is expected to be accurate in areas with higher borehole densities; however, it may have more uncertainty in areas with sparsely located boreholes or areas where subsurface heterogeneity exists at length scales smaller than the borehole separation.

## 3. Methods

### 3.1. AEM system and data acquisition





At the Hanford Site 600 Area, the eastern portion of the Hanford Site, a Fugro (now CGG) RESOLVE® system was used to collect AEM data. The objective of this AEM survey was to delineate stratigraphic/lithologic changes in the upper layers to a depth of approximately 50 m.

The RESOLVE® system is based on the frequency-domain AEM method and measures the secondary EM field induced in the ground by a transmitter loop. It operates with five pairs of horizontal co-planar (HCP)
loop-loop configurations at 400, 1800, 8200, 40000, and 140000 Hz nominal frequencies, and one pair of vertical co-axial (VCX) loop-loop configuration at 3300 Hz. The transmitter-receiver separation is 7.9 m for the HCP coil sets and 9.0 m for the VCX coil set. The transmitter and receiver coil sets are mounted in a "bird", which is towed beneath an aircraft or helicopter during data acquisition.

The blue enclosed area, adjacent to the Columbia River, in Fig. 1 shows the AEM survey area at the Hanford
Site 600 Area. The RESOLVE® system was towed beneath a helicopter and flown at an average speed of 133 km/h and a nominal ground clearance of 30 m over the survey area between June 29 and July 1, 2008. In some portions of the survey area, e.g., above power lines, the ground clearance was much higher than the nominal ground clearance due to safety reasons. The actual operational frequencies were 385, 1792, 8180, 41060, and 128520 Hz for the five pairs of the HCP configurations, and 3342 Hz for the VCX
configuration. The survey was distributed along 29 flight lines (6 lines L10010–L10060, 6 lines L10011–L10061, 9 lines L10070–L10150, and 8 lines L10300–L10370) oriented in a south-to-north direction with a spacing of about 200 m between flight lines, except for the eight easternmost lines L10300–L10370 where a flight-line separation of about 100 m was used. The orientation of these flight lines over the survey area is shown in Fig. 3. Most of the flight lines pass over two power lines, P-1 and P-2 (red lines in Fig. 3).
Measurements of the AEM data were made approximately every 3.7 m along the flight line, which resulted in about 105,000 AEM sounding locations for 412 line-km of the survey. With 12 channels of data recording at each sounding, both in-phase and quadrature components for six frequencies, the survey resulted in about 1.26 million data points.

The survey area has 22 boreholes (B-1 to B-22) within or nearby (depicted with green dots in Fig. 3). Of
these, 16 boreholes are located directly below the AEM survey flight lines: 7 boreholes (B-15 to B-21) below lines L10300–L10370, 8 boreholes (B-3/4/6, B-8 to B-11, and B-14) below lines L10011–L10061, 1 borehole (B-12) below lines L10010–L10060. There are no boreholes located below lines L10070–L10150. The 3D GFM, built in this area by interpolating depths of borehole geologic contacts, is expected to be a good approximation below lines L10300–L10370, where boreholes provide reasonably good
coverage. However, uncertainty in the GFM increases below lines L10070–L10150 and L10010–L10060



due to the sparse borehole coverage. The GFM may also be uncertain below lines L10011–L10061, where boreholes are mostly clustered near one location.

### 3.2. AEM data modeling and inversion

The forward modeling presented in this paper is performed using a 3D unstructured-mesh finite-element forward modeling software. Assuming a temporal-dependence of $e^{-i\omega t}$ with the angular frequency $\omega$ and $i = \sqrt{-1}$, a quasi-static vector Helmholtz equation for the electric field can be derived from Maxwell's equations. Following Jaysaval et al. (2014, 2015, 2016), we have

$$\nabla \times \nabla \times \mathbf{e} - i\omega\mu\sigma\, \mathbf{e} = i\omega\mu\, \mathbf{j}\,, \tag{1}$$

where $\mathbf{e}$ is the intensity of the electric field; $\mathbf{j}$ is the electric field source; and $\mu$ and $\sigma$ are, respectively, the magnetic permeability and electrical conductivity of the medium. The electrical conductivity is inverse of the electrical resistivity $\rho$.

To get accurate results for a frequency-domain AEM system, we solve Eq. (1) by decomposing total electric field $\mathbf{e}$ into primary $\mathbf{e}_p$ and secondary $\mathbf{e}_s$ fields (Newman and Alumbaugh, 1995) as

$$\mathbf{e} = \mathbf{e}_p + \mathbf{e}_s \tag{2}$$

and

$$\Delta\sigma = \sigma - \sigma_p\,, \tag{3}$$

where $\sigma_p$ is the background conductivity of the medium. Using Eqs. (1), (2), and (3), and assuming that the primary field $\mathbf{e}_p$ satisfies Eq. (1), we get

$$\nabla \times \nabla \times \mathbf{e}_s - i\omega\mu\sigma\, \mathbf{e}_s = i\omega\mu\Delta\sigma\, \mathbf{e}_p\,. \tag{4}$$

The edge based finite element discretization of Eq. (4) on a tetrahedral mesh is assembled into a system of linear equations (Jin, 2002):





$$\mathbf{M}\mathbf{x} = \mathbf{s} \,, \tag{5}$$

where $\mathbf{M}$ is the system matrix, $\mathbf{x}$ is the vector of unknown electric field, and $\mathbf{s}$ is the source vector resulting from the right-hand side of Eq. (4). This equation is solved to compute the electric field followed by using Faraday's law to compute the magnetic field. Accuracy of the developed forward modeling software was benchmarked against analytical/semi-analytical solutions for layered earth and Jaysaval et al. (2014, 2015, 2016) for 3D earth models. The results agreed well, within acceptable error ranges, depending on fine- or coarse-meshing of benchmarking models.

In the inversion, we have implemented a Gauss-Newton optimization scheme following Johnson et al. (2010). The objective of the inversion is to find a conductivity model $\mathbf{m}$ (in our case it represents logarithmic of the conductivity) such that it minimizes a cost function $\phi$:

$$\phi = \phi_\mathrm{d} + \beta \phi_\mathrm{m} \,, \tag{6}$$

where $\phi_\mathrm{d}$ is the data cost function measuring the misfit between the observed $d_\mathrm{obs}$ and forward modeled $d_\mathrm{syn}$ data; $\phi_\mathrm{m}$ is the corresponding misfit measure between $\mathbf{m}$ and constraints placed upon it; and $\beta$ is a regularization parameter controlling the contribution of $\phi_\mathrm{m}$ to $\phi$ compared to $\phi_\mathrm{d}$. We choose to minimize $\phi$ using the least-square method, and hence the data and model cost functions are, respectively,

$$\phi_\mathrm{d} = \left\| \mathbf{W}_\mathrm{d} \{ \mathbf{d}_\mathrm{obs} - \mathbf{d}_\mathrm{syn}(\mathbf{m}) \} \right\| \tag{7}$$

and

$$\phi_\mathrm{m} = \left\| \mathbf{W}_\mathrm{m}(\mathbf{m} - \mathbf{m}_\mathrm{ref}) \right\| . \tag{8}$$

Here, $\mathbf{m}_\mathrm{ref}$ is the logarithmic of a reference conductivity model, $\mathbf{W}_\mathrm{d}$ is the data-weighing matrix, and $\mathbf{W}_\mathrm{m}$ is the regularization matrix.

We set $\partial \phi / \partial \mathbf{m} = 0$ to minimize $\phi$ using the least-square method. This results in the following normal equation:



$$[Re\{\mathbf{\mathcal{J}}^{\mathrm{H}}\mathbf{W}_{\mathrm{d}}^{\mathrm{T}}\mathbf{W}_{\mathrm{d}}\mathbf{\mathcal{J}}\} + \beta\mathbf{W}_{\mathrm{m}}^{\mathrm{T}}\mathbf{W}_{\mathrm{m}}]\delta\mathbf{m}$$
$$= Re[\mathbf{\mathcal{J}}^{\mathrm{H}}\mathbf{W}_{\mathrm{d}}^{\mathrm{T}}\mathbf{W}_{\mathrm{d}}\{\mathbf{d}_{\mathrm{obs}} - \mathbf{d}_{\mathrm{syn}}(\mathbf{m})\}] - \beta\mathbf{W}_{\mathrm{m}}^{\mathrm{T}}\mathbf{W}_{\mathrm{m}}(\mathbf{m} - \mathbf{m}_{\mathrm{ref}}),$$

(9)

where $\mathbf{\mathcal{J}}$ is the Jacobian or sensitivity matrix that represents partial derivatives of the synthetic data with respect to the model parameters and $\delta\mathbf{m}$ is a model update vector. The superscripts T and H denote, respectively, the transpose and conjugate transpose operators. The inversion is performed iteratively: at each iteration with a current model $\mathbf{m}^{k}$, we seek to obtain $\delta\mathbf{m}^{k}$ by solving normal Eq. (9) such that a new

model

$$\mathbf{m}^{k+1} = \mathbf{m}^{k} + \alpha\,\delta\mathbf{m}^{k}$$

(10)

decreases the cost function $\phi$. Here, $\alpha$ is the step length. This iterative process continues until the data misfit reaches below a pre-defined tolerance level.

The Jacobian matrix is computed using the adjoint-state method (McGillivray et al., 1994), and the normal equation (Eq. 9) is solved using a conjugate-gradient least-squares method following Johnson et al. (2010).

Further implementation details of the Jacobian matrix and the solver are beyond the scope of the current paper. However, we do emphasize that they are massively parallelized and usually computed using several thousands of processing cores for large-scale 3D problems.

### 3.3. Electrical resistivity and lithology

The application of any geophysical EM or electrical method is primarily based on the relationship of

electrical resistivity with lithology, porosity, and/or pore fluids (Palacky, 1987). The knowledge of such a relationship will guide us to interpret resistivity models obtained by inverting the collected AEM data at the Hanford Site 600 Area.

In sedimentary rocks, there are two dominant mechanisms that contribute to electrical resistivity: ionic conduction and surface conduction (Knight and Endres, 2005). The ionic conduction occurs through pore

fluids and its efficiency depends on porosity, pore-fluid saturation, and pore-fluid type, e.g., in sedimentary rocks with water in pore space, the resistivity tends to decrease as water saturation increases. As a result, fully saturated sedimentary sequences at the Hanford Site will have a lower resistivity value than their less-saturated counterparts. Ionic conduction is also affected by the grain size: the coarser the grain size, the higher will be electrical resistivity because the bulk of the volume is dominated by insulating minerals.



Therefore, gravels and sands of the Hanford formation are expected to have higher resistivity values than adjacent finer-grained sediments of the Ringold Formation.

The second mechanism, the surface conduction, occurs due to the presence of a high concentration of ions associated with the electrical double layer at the solid/water interface (Knight and Endres, 2005). The occurrence of surface conduction causes a decrease in electrical resistivity and its efficiency depends on

the surface area: the higher the surface area, the higher the surface conduction, which is inversely related to the electrical resistivity. As clays and finer grains have higher surface areas, the resistivity tends to decrease as the clay content increases and as the grain size decreases. Consequently, at the Hanford Site, the finer-grained sediments of the Ringold Formation are expected to have lower resistivity values than coarse-grained sediments. For example, sediments in the Ringold Mud unit — zones of increased silt/clay

contents within the Ringold Formation — are expected to have low resistivity values.

In igneous rocks, the resistivity depends on their silica content: the more silica the rock matrix contains, the more resistive the rock becomes (Palacky, 1987). Although basalt has relatively less silica content compared to other igneous rocks, e.g., felsic igneous rocks, its resistivity is usually much higher than common sedimentary rocks and can vary in the order of 1000 Ωm.

## 4.   Results and discussion

### 4.1.   Synthetic study for full 3D inversion

We first performed a synthetic inversion study to determine if a full 3D inversion of the AEM data along parallel flight lines with 200 m of flight-line separation could recover the subsurface resistivity at the Hanford Site. For this, we considered a four-layer earth model, shown in Fig. 4a. The resistivities of the

layers from top to bottom are 1000, 3000, 600, and 50 Ωm, respectively, with the corresponding thicknesses of 35, 25, 20, and 20 m. The dimension of the model is $1000 \times 700 \times 100\ m^3$. Excluding the top side, each side of the model is padded with 3 km to accommodate zero-field or Dirichlet boundary conditions. The top side includes a highly resistive air layer of resistivity $10^7$ Ωm and thickness 10 km.

For this model, we computed synthetic AEM data at 30 m height from the ground surface for 385, 1792,

8180, 41060, and 128520 Hz frequencies simulating the five pairs of the HCP configurations in the RESOLVE® system. These data were computed at every 20 m along a flight line for three different cases of the flight-line separation. In the first case, the flight lines were separated by 200 m and located at $x = 50, 250, 450,$ and 650 m along the $y$-direction. In the second case, the flight lines were separated by 100 m and located at $x = 50, 150, 250, 350, 450, 550,$ and 650 m along the $y$-direction. Similarly, in the third



case, the flight lines were separated by 50 m and located from $x = 50$ m to $x = 650$ m at every 50 m along
the $y$-direction.

The simulated AEM data were inverted using the previously described inversion algorithm using a starting
model having half-space resistivity of 500 Ωm. The inversion iterations were stopped after the data misfit
reduced to 1% of the initial data misfit value in all the three cases. Figs. 4b, 4c, and 4d, respectively, show

the inverted resistivity models for the three cases of 200, 100, and 50 m flight-line separations. The
inversion of the synthetic AEM data for 200 m flight-line separation (Fig. 4b) could not recover the true
resistivity model of Fig. 4a. The model failed to recover the resistivity of the upper two layers and
incorrectly produced highly resistive artifacts in the top layer. This implies that 200 m flight-line separation
was too large to resolve resistivity structures in between flight lines. On the other hand, the inversion

accurately recovered the true resistivity model for the AEM data with 50 m flight-line separation (see Fig.
4d). For 100 m flight-line separation (Fig. 4c), the inversion recovered the true resistivity, but produced
some artifacts, shown as intervals of highly resistive features in the top layer.

### 4.2.    Inversion of AEM data at the Hanford Site 600 Area

In the previous section, we demonstrated that 200 m flight-line separation was too large for full 3D inversion

to recover the true resistivity model. As the AEM data at the Hanford Site 600 Area were predominantly
collected with about 200 m flight-line separation, we inverted the data along individual flight lines. This is
equivalent to resolving a 2D resistivity model along each flight line, although the inversions were performed
considering the full 3D forward modeling.  The AEM data were inverted at every fifth sounding location
to lessen data redundancy. This resulted in about 18.5 m spacing between sounding locations along the

flight path. Moreover, we only inverted the HCP channels, both in-phase and quadrature, for five pairs with
frequencies, respectively, 385, 1792, 8180, 41060, and 128520 Hz. The VCX channels were not inverted
because our implementation of the Gauss-Newton inversion algorithm did not yet allow for mixing the HCP
and VCX data. The starting model was a 200 Ωm half-space for each inversion along an individual flight
line. The inversions were allowed to converge when the data misfit reduced to about 5% of the initial data

misfit value. Plots comparing the in-phase and quadrature components of the observed and predicted data
for the inverted resistivity models are shown in Fig. 5 for all five frequencies. These plots suggest that the
predicted data are qualitatively agreeing well with the observed ones.

We now show the inverted resistivity model along flight-line L10300 (see Fig. 3 for the line orientation) in
Fig. 6. The model is shown down to 100 m depth. However, the result may be reliable only up to about 60

m depth, which is the effective penetration depth limit of the RESOLVE® AEM system reported by the



operator (Fugro). Compared to other flight lines, this line has the most boreholes nearby with good coverage; it has three boreholes, B-15, B-16, and B-17 (shown by green dots near line L10300 in Fig. 3 and by black vertical lines in Fig. 6), which helps to validate the inversion result. Three geologic contacts, the top of the Ringold E unit, the Ringold Mud unit, and the Columbia River Basalt Group (hereinafter the

basalt), extracted from the GFM are overlain on this resistivity model and shown, respectively, by black, magenta, and orange curves in Fig. 6. These three contacts were also interpreted from boreholes B-15, B-16, and B-17, and are marked, respectively, by black, magenta, and orange cross marks. We notice a minor mismatch between some of the geologic contacts derived from the boreholes and GFM. This is because the boreholes may not be located exactly below the flight line. Two power lines passing this cross-section are

marked by P-1 and P-2. These power lines yielded wide swaths of unreliable AEM data, and therefore inverted resistivity values near the power lines should be disregarded for any interpretation for this flight line as well as all the other flight lines passing over these power lines.

The inverted resistivity model shows a general trend of high-to-low relative change in resistivity values with increasing depth. The zone with high resistivity values ($\rho > 400$ Ωm) is observed in the uppermost

layer and correlates very well with the Hanford formation interpreted from the GFM and three neighboring boreholes B-15, B-16, and B-17, as shown above the black curve and black cross marks in Fig. 6. The Hanford-Ringold E contact seems to correlate with a change in resistivity from greater than 400 Ωm for the Hanford formation to less than 300 Ωm for the Ringold E unit. The layer of relatively low resistivity ($< 80$ Ωm) at depth probably correlates with the Ringold Mud unit or with units of increased clay/silt within the

Ringold Formation. The Ringold E-Ringold Mud contact also shows fairly good correlation with a change in resistivity from greater than 100 Ωm for the Ringold E unit to less than 80 Ωm for the Ringold Mud unit, although this contact is not observed in the southernmost part of the inverted resistivity model. The bottom basalt layer (shown below 50 m depth) is expected to have very high resistivity values. However, the inversion of the AEM data could not map this resistive layer, possibly due to the penetration depth limit of

the RESOLVE® system. In addition, even though the geological contacts derived from GFM correlate well with the inverted AEM model, there are some discrepancies, e.g., the Ringold E-Ringold Mud contact in the northernmost part (Northing: 130000–133000), possibly due to the 3D GFM interpretation or lower penetration depth of the AEM data. Another explanation for the intra-Ringold E change from high resistivity ($> 100$ Ωm) to low resistivity ($< 80$ Ωm) could be a change in coarse- to finer-grained Ringold

sediments or clayey intervals within the Ringold E unit.

In the following inversion results for all other flight lines, we used the above interpretation of resistivity values for characterizing the stratigraphic units. The Hanford formation is characterized by high resistivity



values ($\rho > 400$ $\Omega$m), the Ringold E unit by intermediate resistivity values ($80$ $\Omega$m $\leq \rho \leq 400$ $\Omega$m), and the Ringold Mud unit by low resistivity values ($\rho < 80$ $\Omega$m). This is summarized in Table 1.

Figs. 7a–7g show the inverted resistivity models, respectively, along flight-lines L10370–L10310. There are four boreholes B-18 to B-21 below these flight lines, which are shown on the inverted models depending on their proximity to the lines, e.g., B-20 and B-21 are located close to line L10370 and therefore are shown on the inverted model along this line. Two power lines, P-1 and P-2, are also marked on the inversion results. As in Fig. 6, the tops of the Ringold E, Ringold Mud, and basalt are shown, respectively, by black,
magenta, and orange curves and cross marks, respectively, obtained from the GFM and boreholes (if available in the vicinity). Similar to the flight-line L10300 result, the inverted resistivity values (see Table 1) for lines L10370–L10310 correlate fairly well with the geological contacts derived from GFM and boreholes. Note that flight-lines L10300–L10370 have reasonably good coverage of boreholes; therefore, the interpolated GFM below these flight lines is expected to be less uncertain.

Figs. 8a–8i present the inverted resistivity models, respectively, along flight-lines L10150–L10070. As done previously, the geologic contacts and power lines are shown on these inverted models. Below these flight lines, there is no borehole coverage; therefore, the GFM may be more uncertain as compared to those shown in Figs. 6 and 7. The Hanford-Ringold E contact derived from the GFM correlates well with the inverted resistivity model (see Table 1) for the easternmost flight-line L10150. However, as we move
westward to line L10070 from line L10150, this contact appears to go deeper in the inverted resistivity models than the one derived from the GFM. Nonetheless, the Ringold E-Ringold Mud contact derived from the GFM seems to agree reasonably well with the inverted resistivity models.

       Figs. 9a–9f and 9g–9l show the inverted resistivity models, respectively, along flight-lines L10061–L10011 and L10060–L10010. The geologic contacts, power lines, and adjacent boreholes are again shown
(wherever present) on these inverted models. Despite containing eight boreholes beneath, flight-lines L10061–L10011 have poor borehole coverage because most boreholes are clustered near one location (see Fig. 3). Flight-lines L10060–L10010 are covered by only a single borehole. Therefore, the GFM uncertainty below these lines is higher. In Figs. 9a, 9b, 9d, 9e, and 9l, the Hanford-Ringold E contact interpreted from the inversion results (see Table 1) shows fairly good correlation with the contact derived from boreholes
B-14, B-14, B-8/9/10/11, B-3/4/6, and B-1, respectively. However, the inversion result for the Hanford-Ringold E contact appears to be deeper relative to the borehole B-12 derived contact (see Fig. 9i). Nevertheless, B-12 is located very close to power line P-1 and, as mentioned previously, the inverted resistivity values should not be considered for the interpretation. Generally, for these flight lines, the geologic contacts derived from GFM show poor agreements with the ones interpreted from the inverted



models, specifically, the Hanford-Ringold E contact, which is consistently deeper in the inverted model than the one derived from GFM.

In all the inversion results, we notice some intervals of thickening of resistive zones (of the Hanford formation) within the Ringold E unit. These may represent intra-Ringold E facies changes from finer- to coarser-grained intervals. An alternate explanation is that these changes may possibly correspond to the

actual Hanford sediments filled into channels incised into the top of the Ringold E unit. The Hanford formation contains coarse-grained sediments that act as transmissive pathways to the hazardous contaminants within the groundwater, and therefore can shorten travel times to downgradient recipients (e.g., the Columbia River). The delineation of these zones forms the technical basis for monitoring and remediation plans for contaminants at the Hanford Site.

**5. Conclusions**

We have successfully inverted 412 line-km of frequency-domain AEM data at the Hanford Site with a massively parallel 3D EM modeling and inversion code. The inversion results delineated subsurface stratigraphic units at the Hanford Site. In areas with good coverage of boreholes, the results showed a good match with an existing GFM built by interpolating geologic contact depths derived from these boreholes.

Outside these areas, where the framework model is more uncertain, the inversion results provided additional information on geologic contact locations for the stratigraphic units. This knowledge can further be incorporated into the GFM to reduce its uncertainty and provide a technical basis for monitoring and remediation activities.

*Author contributions*. JLR and PJ planned the study. PJ implemented the modeling and inversion code,

performed the inversion, analyzed the results, and made the figures. JLR contributed to analyze the results and to make figures for the location map. TCJ was supervising the study. PJ wrote the manuscript, and all co-authors read the manuscript and helped with editing.

*Competing interests*. The authors declare that they have no conflict of interest.

*Acknowledgements*. The authors thank Mark L. Rockhold for his support in extracting the geologic

framework model beneath the survey area, and Rob D. Mackley for discussion on inversion results and the stratigraphy of the Hanford Site. We also express our gratitude to Sarah D. Springer for providing the stratigraphic contact depths interpreted from 22 boreholes in/near the survey area.



*Financial support.* This research was supported by the U.S. Department of Energy Richland Operations Office under the Deep Vadose Zone - Applied Field Research Initiative at Pacific Northwest National Laboratory. The Pacific Northwest National Laboratory is operated by Battelle Memorial Institute for the U.S. Department of Energy under Contract DE-AC05-76RL01830.

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



**Table 1:** Interpretation of the inverted resistivity values for characterizing the stratigraphic units at the Hanford Site.

| Inverted Resistivity Values (Ωm) | Interpreted Formation/Unit |
|---|---|
| $\rho > 400$ | Hanford |
| $80 \leq \rho \leq 400$ | Ringold E |
| $\rho < 80$ | Ringold Mud |



**Figure 1:** Location map showing the survey area (enclosed in blue) at the Hanford Site, located in southeastern Washington State. The enclosed area with dashed red line shows the Central Plateau. Base map © Google, LLC.



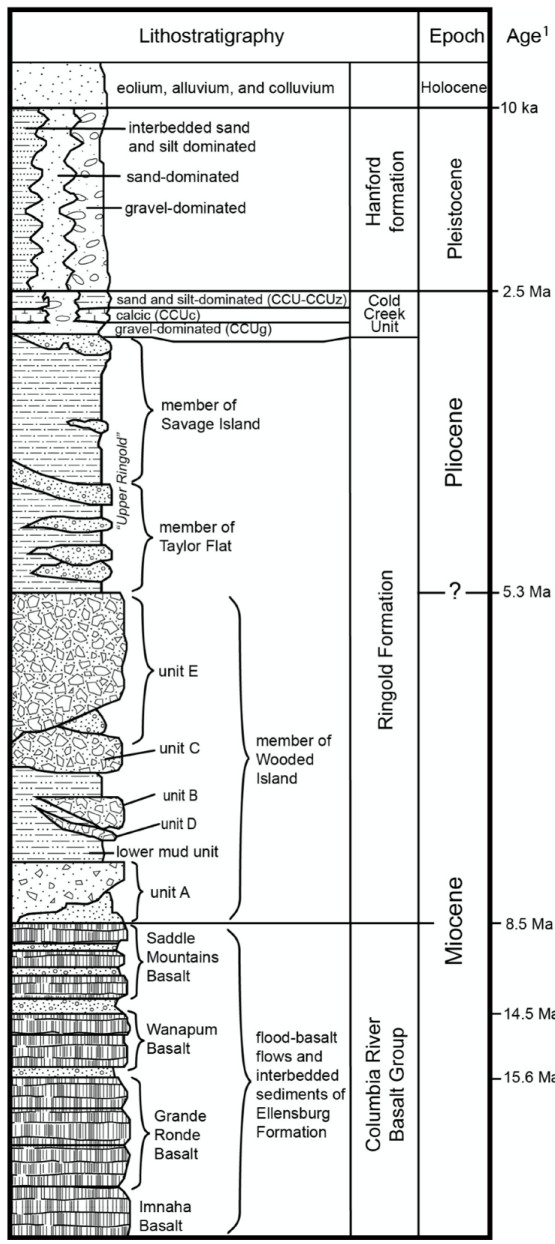

**Figure 2:** Layer-cake depositional model of the generalized stratigraphy at the Hanford Site. Sources: DOE (2002), Lindsey (1995), and Lindsey et al. (1994). (Not to scale)

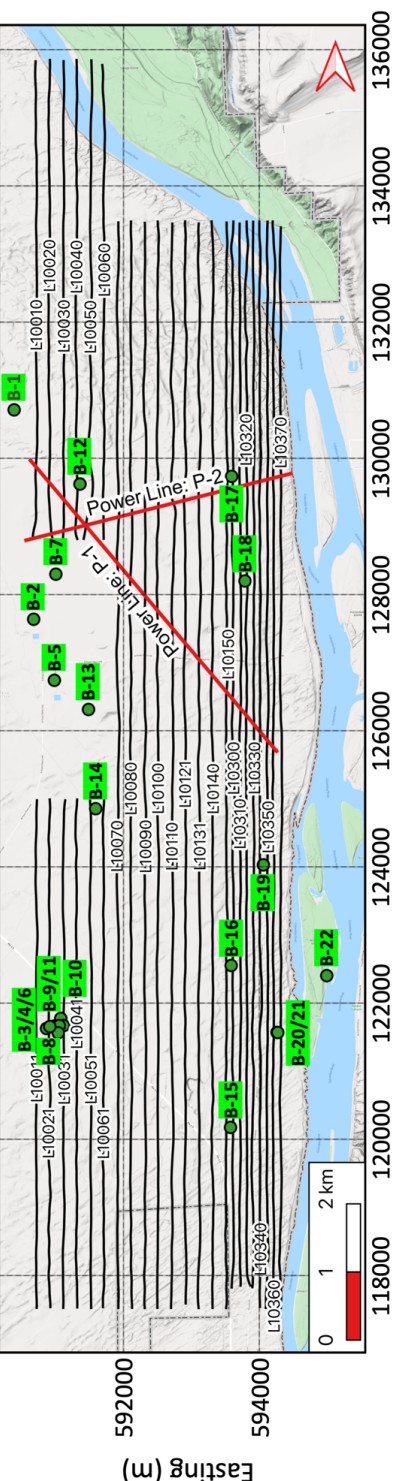

**Figure 3:** Location map showing the AEM flight lines (black lines), boreholes (green dots), and power lines (red lines). A total of 29 AEM flight lines were used and are numbered as L10010–L10060, L10011–L10061, L10070–L10150, and L10300–L10370. The survey area includes 22 boreholes within or nearby, which are numbered as B-1 to B-22. Relative location of the survey area is shown in Fig. 1. Base map © Google, LLC.

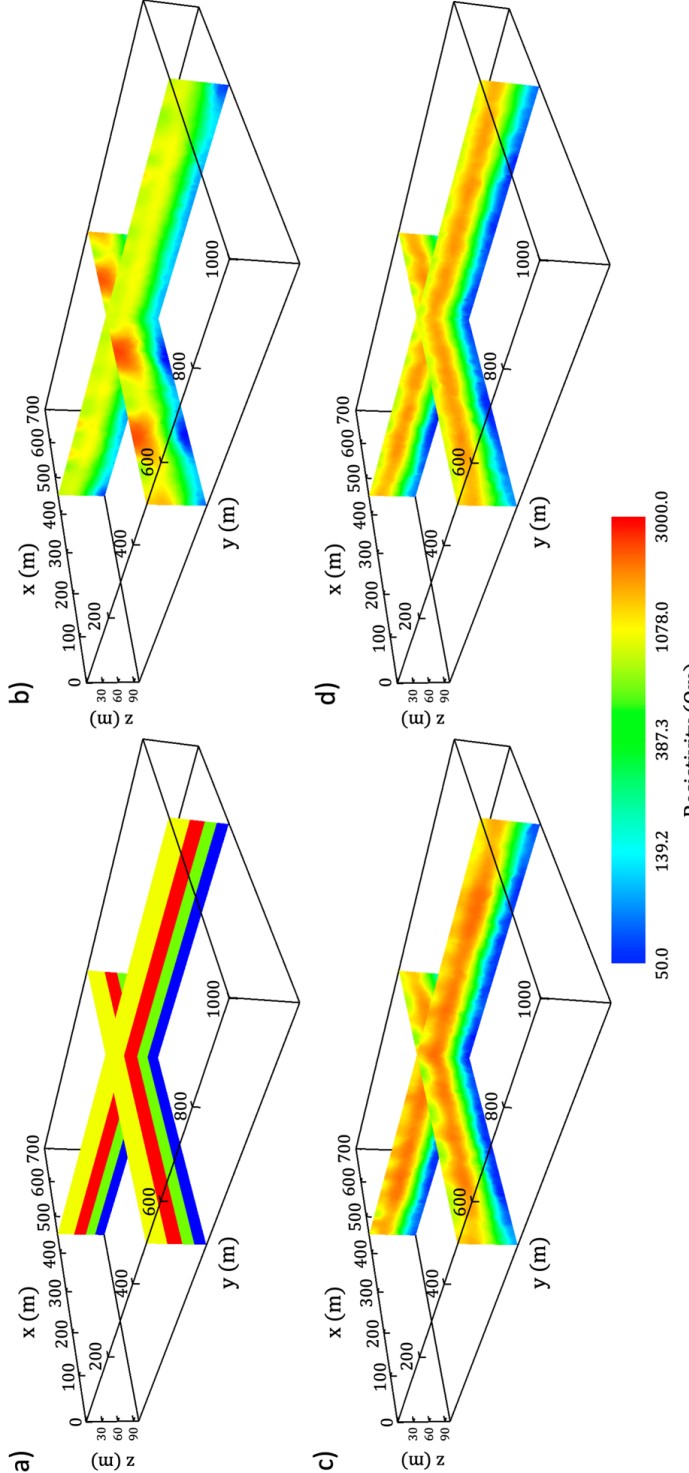

**Figure 4:** True resistivity model and inversion results for the synthetic study: **(a)** True four-layer earth model, **(b)**, **(c)**, and **(d)** inverted resistivity models for three different cases, respectively, with 200, 100, and 50 m flight-line separations.

595





**Figure 5:** Observed and predicted data (for inverted models) for the AEM survey at the Hanford Site. Top panels: in-phase and bottom panels: quadrature components, for five HCP pairs with frequencies, respectively, of **(a)** 385, **(b)** 1792, **(c)** 8180, **(d)** 41060, and **(e)** 128520 Hz.



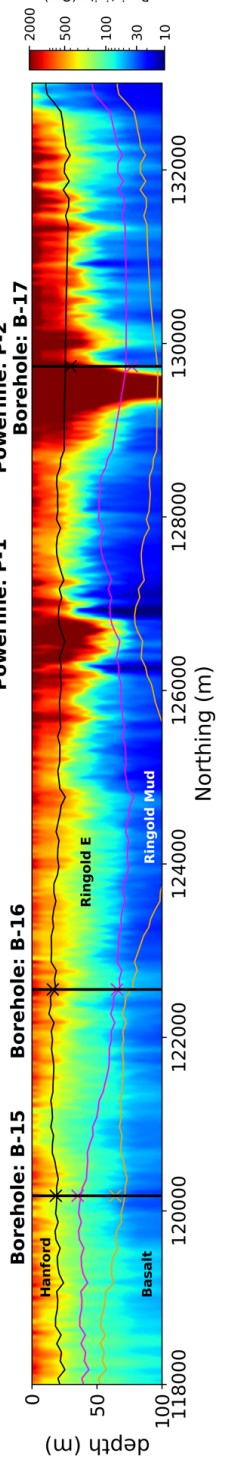

**Figure 6:** AEM data inversion result along flight-line L10300. Three boreholes, B-15, B-16, and B-17 (vertical black lines), are located near this flight line. Two power lines, P-1 and P-2, pass below this flight line. Overlain this result are three geological contacts, the top of Ringold E, Ringold Mud, and basalt, shown by black, magenta, and orange curves and crosses, respectively, derived from the geological framework model (GFM) and boreholes. (Vertical exaggeration or VE = 15)

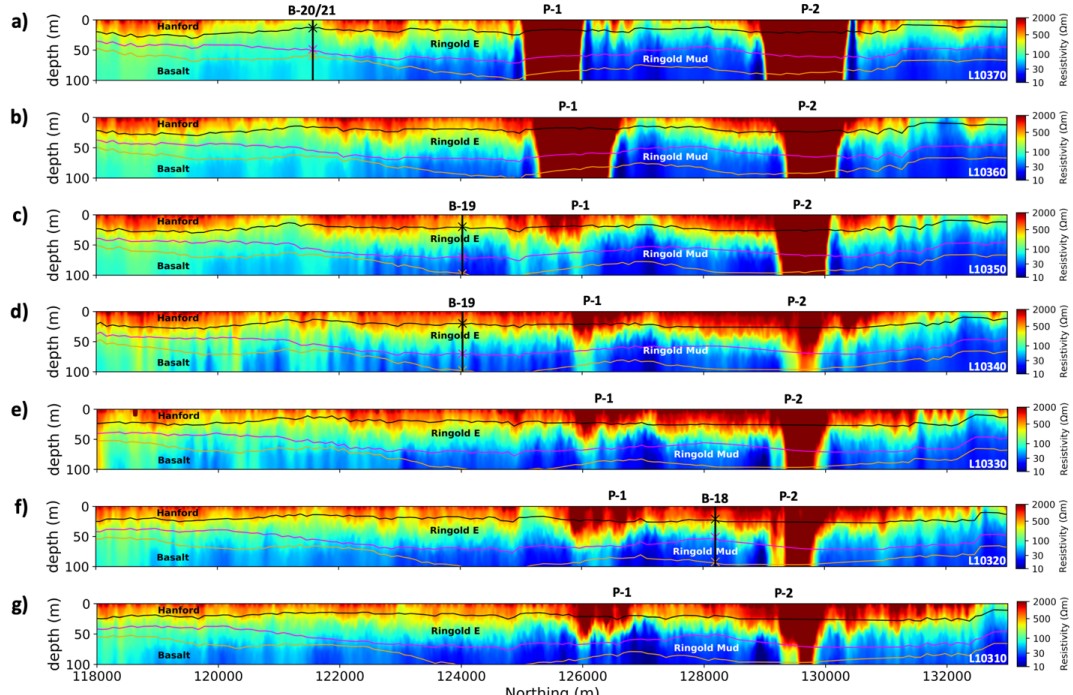

**Figure 7:** AEM data inversion results along flight-lines **(a)** L10370, **(b)** L10360, **(c)** L10350, **(d)** L10340, **(e)** L10330, **(f)** L10320, and **(g)** L10310. Four boreholes, B-18, B-19, B-20, and B-21, are placed on these results depending on their proximity to these flight lines (vertical black lines). Two power lines, P-1 and P-2, pass below these lines. Overlain these results are three geological contacts, the top of Ringold E, Ringold Mud, and basalt, shown by black, magenta, and orange curves and crosses, respectively, derived from the GFM and boreholes. (VE = 10)



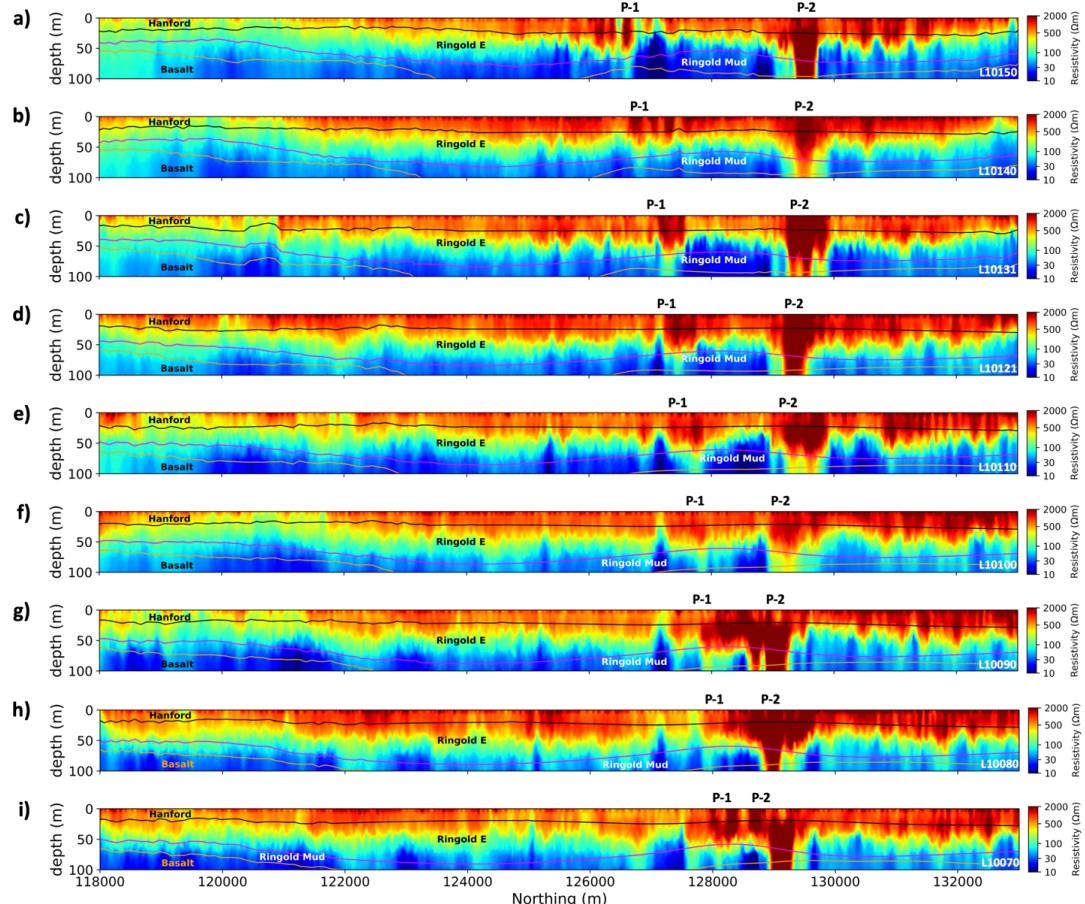

650

**Figure 8:** AEM data inversion results along flight-lines **(a)** L10150, **(b)** L10140, **(c)** L10131, **(d)** L10121, **(e)** L10110, **(f)** L10100, **(g)** L10090, **(h)** L10080, and **(i)** L10070. Two power lines, P-1 and P-2, pass below these lines. Overlain these results are three GFM-derived geological contacts, the top of Ringold E, Ringold Mud, and basalt, shown, respectively, by black, magenta, and orange curves. (VE = 10)

655



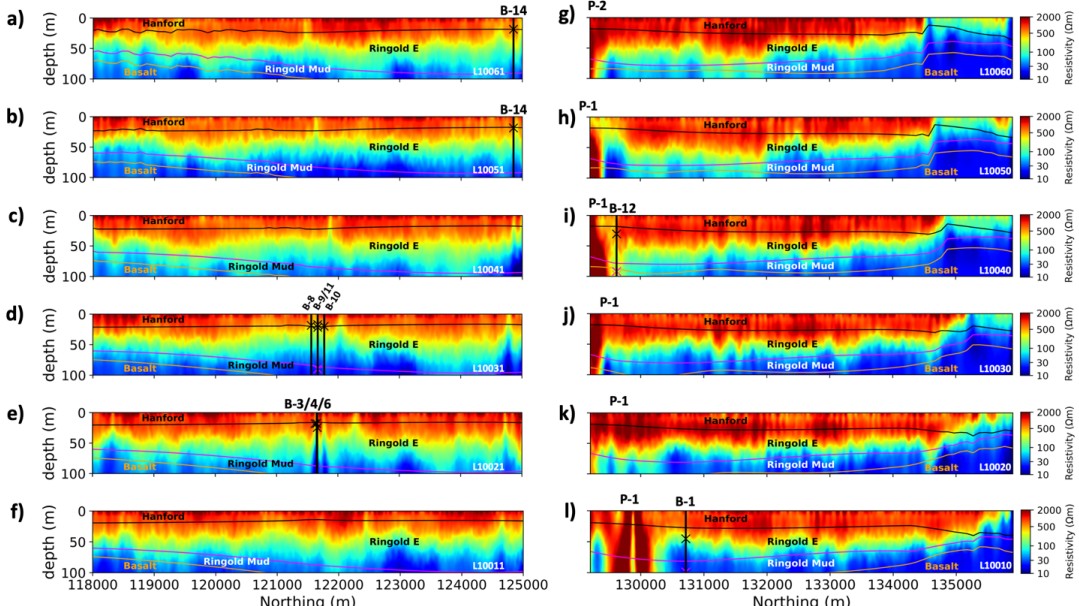

**Figure 9:** AEM data inversion results along flight-lines **(a)** L10061, **(b)** L10051, **(c)** L10041, **(d)** L10031, **(e)** L10021, **(f)** L10011, **(g)** L10060, **(h)** L10050, **(i)** L10040, **(j)** L10030, **(k)** L10020, and **(l)** L10010. Boreholes B-3/4/6, B-8 to B-12, and B-14 are placed on these results depending on their proximity to these flight lines (vertical black lines). Two power lines, P-1 and P-2, pass below some of these lines. Overlain these results are three geological contacts, the top of Ringold E, Ringold Mud, and basalt, shown by black, magenta, and orange curves and crosses, respectively, derived from the GFM and boreholes. (VE = 10)