# Peer review of "Stratigraphic Identification with Airborne Electromagnetic Methods at the Hanford Site, Washington"

_Hydrology and Earth System Sciences, 2020_

## Referee Comment (RC1) · Anonymous Referee #1 · 8 Nov 2020

This mansucript presents a nice application of 3D inversion of Airborne EM data. The goal of the work is strictly related to the assessment of the applicability of inversion codes to a large data set of EM data, as explicitly declared by the authors: "in this paper, we present the first inversion results of the AEM data at the Hanford Site obtained using a 3D EM modeling and inversion code" (lines 79-80).

The work is well done, well written and interesting from the point of view of geophysical prospecting. However, it does not provide a new methodological approach, as it is largely based on the results of previous papers by the first author. The novelty is focused on the application of the methods developed there to a large data set. However,

the work does not include any discussion of hydrological process, of an hydrological model or hydrological data.

Therefore, I think that the work is very far from the goals of HESS and should be submitted to one of the journals which accept papers strictly dealing with geophysical exploration (e.g., Geophysics, Geophysical prospecting, Journal of applied geophysics, Near surface geophysics, etc.).

---

## Referee Comment (RC2) · Anonymous Referee #2 · 23 Nov 2020

The paper presents a 3D forward modeling inserted in 2D inversion schemes of an Airbone electromagnetic (AEM) data set. The data set is consequent and the computation is performed thanks to an extensive parallelization of the code. The results of the inversion are compared with a geologic framework model (GFM) built from boreholes data. AEM images show divergences with the GFM where it is more uncertain due to a lack of borehole. The resistivity images presented here are a good proxy to delineate the interfaces between the different geological units of the studied area. Some geological units being more transmissive than others, the knowledge of their respective pattern is necessary to estimate the transport of the contaminants that have been discharged on the studied site.

[Figure]

The paper is very clear, well written and organized. However, some information are lacking to evaluate the accuracy of the proposed forward model. Also to be published in a journal such HESS, authors should develop their results beyond a qualitative comparison with a GFM. For instance, a probabilistic study could be applied similarly as in the paper of Gottschalk et al., (2017). Then, an estimate of the influence of the information given by the AEM data inversion on the contaminant transport with respect to an hydrogeological model delimited from the GFM only would enhance the interest of the work presented here. So substantial work is required to publish this paper in HESS.

Gottschalk, I. P., Hermans, T., Knight, R., Caers, J., Cameron, D. A., Regnery, J., & McCray, J. E. (2017). Integrating non-colocated well and geophysical data to capture subsurface heterogeneity at an aquifer recharge and recovery site. Journal of Hydrology, 555, 407-419.

Below are detailed comments: Line 75: "Both ERT and ground-based EM data were inverted using 2D inversion schemes and the obtained resistivity models were fairly consistent with the stratigraphic units interpreted from six nearby borehole datasets down to 30–40 m depth. However, these inversion results showed poor agreement with the resistivity slices computed from the AEM data." → How is the agreement of the AEM result of inversion presented here with the ERT and ground-based EM data inversion ?

Line 200: "Accuracy of the developed forward modeling software was benchmarked against analytical/semi-analytical solutions for layered earth and Jaysaval et al. (2014, 2015, 2016) for 3D earth models. The results agreed well, within acceptable error ranges, depending on fine- or coarse-meshing of benchmarking models." → None of these results are presented here so it is impossible to me to validate the code accuracy.

Line 286: "This is equivalent to resolving a 2D resistivity model along each flight line, although the inversions were performed considering the full 3D forward modeling." →

This test should be considered in the different synthetic cases explored to show how well it resolves the underground structures. Also synthetic cases could study the difficulty to map the bottom interface of a deep resistive structure located below a conductive one... The electrical current lines being trapped in the conductive layer it is well known that the accuracy of this boundary is hard to evaluate, all the more if it is located close to the limit of the tool penetration depth.

The way the results are presented is a bit disappointing as it is very qualitative while a deeper interpretation could better show the interest of your work. First instead of showing all the resulting images by profile you could show it with 2D maps of the targeted interfaces depths compared with the same maps given by the GFM. Boreholes should be placed on such maps. Moreover, since the power lines perturb the AEM images, region with unreliable results should be blanked.

Regions with a higher uncertainty of the GFM should be distinguished: a 2D map of the interfaces location likelihood should be presented.

---

## Author Comment (AC1) · 8 Dec 2020

We thank you for your feedbacks. We submitted the manuscript to HESS thinking it would be useful to the hydrology community. We agree that the manuscript doesn't show a direct application to hydrologic processes/models/data, but we believe it does have a significant relevance to hydrogeology. However, following your suggestion and further conversation with the editor, we will withdraw the manuscript from HESS and submit to a journal strictly dealing with geophysical exploration. Thank you.